

# Validation for the function of protein C in mouse models

Ya Liu[1,*], Maoping Cai[1,*], Yan Chen[1], Guocai Wu[2], Songyu Li[1] and Zhanghui Chen[1]

[1] Zhanjiang Institute of Clinical Medicine, Central People's Hospital of Zhanjiang, Guangdong Medical University, Zhanjiang, Guangdong, China
[2] Department of Hematology, Central People's Hospital of Zhanjiang, Guangdong Medical University, Zhanjiang, Guangdong, China
[*] These authors contributed equally to this work.

## ABSTRACT

**Objectives.** Protein C (PC) is an anticoagulant that is encoded by the PROC gene. Validation for the function of PC was carried out in mouse models.

**Methods.** In this study, autosomal recessive PC deficiency (PCD) was selected as the target, and the specific mutation site was chromosome 2 2q13-q14, PROC c.1198G>A (p.Gly400Ser) which targets G399S (GGT to AGC) in mouse models. To investigate the role of hereditary PC in mice models, we used CRISPR/Cas9 gene editing technology to create a mouse model with a genetic PCD mutation.

**Results.** The two F0 generation positive mice produced using the CRISPR/Cas9 gene editing technique were chimeras, and the mice in F1 and F2 generations were heterozygous. There was no phenotype of spontaneous bleeding or thrombosis in the heterozygous mice, but some of them were blind. Blood routine results showed no significant difference between the heterozygous mice and wild-type mice ($P > 0.05$). Prothrombin time (PT), activated partial thromboplastin time (APTT), and thrombin time (TT) were prolonged in the heterozygous mice, while the level of fibrinogen content (FIB) decreased, suggesting secondary consumptive coagulation disease. The protein C activity of heterozygous mice was significantly lower than that of wild-type mice ($P < 0.001$), but there was no significant difference in protein C antigen levels ($P > 0.05$). H&E staining showed steatosis and hydrodegeneration in the liver of heterozygous mice. Necrosis and exfoliated epithelial cells could be observed in renal tubule lumen, forming cell or granular tubules. Hemosiderin deposition was found in the spleen along with splenic hemorrhage. Immunohistochemistry demonstrated significant fibrin deposition in the liver, spleen, and kidney of heterozygous mice.

**Conclusion.** In this study, heterozygotes of the mouse model with a PC mutation were obtained. The function of PC was then validated in a mouse model through genotype, phenotype, and PC function analysis.

Corresponding authors
Songyu Li, lsyu@gdmu.edu.cn
Zhanghui Chen, zjcell@126.com, zj-cell@gmail.com

# INTRODUCTION

Venous thromboembolism (VTE) is a complex disease with various factors contributing to its pathogenesis, including genetic factors (*Kim et al., 2014*; *Tang & Hu, 2015*). In Asians,
one of the major genetic factors associated with venous thrombosis is hereditary protein C deficiency (PCD) (*Shen, Lin & Tsay, 1997*; *Kim et al., 2016*; *Kinoshita et al., 2005*). Protein C (PC), a vitamin K-dependent serine protease, is primarily synthesized in the liver. It is encoded by the PC gene (PROC) located on chromosome 2q14.3. The PROC gene spans approximately 11.1 kb and consists of nine exons. Mature PC circulates in plasma, connected by 41kDa heavy chains and 21 kDa light chains through disulfide bonds. Activated PC (APC) occurs through the thrombine-thrombomodulin complex (T-TM). APC acts as an anticoagulant by inactivating coagulation factors Va (FVa) and VIIIa (FVIIIa) *via* proteolysis in the presence of the phospholipid surface, protein S cofactor, and intact factor V (*Sarangi, Lee & Kim, 2010*; *Wildhagen et al., 2011*; *Dahlbäck & Villoutreix, 2005*). PCD can be inherited in an autosomal dominant or recessive pattern. Autosomal recessive PCD, resulting from biallelic PROC gene mutations (homozygote or complex heterozygote), leads to plasma PC levels less than 1% of normal. Homozygous patients are at high risk of fulminant violet paralysis and severe disseminated intravascular coagulation during the neonatal period, which can be fatal if not detected or treated early (*Chalmers et al., 2011*). Heterozygotes with recessive inheritance typically exhibit approximately 50% of normal PC activity, with an incidence rate of about 0.3%. Clinical manifestations of heterozygous individuals include thrombophlebitis, costal vein embolism, cutaneous microvascular embolism, and in some cases, deep vein formation and/or pulmonary embolism (*Cooper, Hill & Maclean, 2012*).

Currently, the primary alternative treatments for hereditary PCD include infusion of fresh frozen plasma, APC concentrate (*Manco-Johnson et al., 2016*; *Hayami et al., 2017*), and the initiation of anticoagulants after bleeding has been controlled. In cases of severe hereditary PCD, liver transplantation is considered the superior treatment (*Boucher et al., 2018*). However, the limited availability of liver transplantation sources and the use of domestic PC concentrates, along with the high cost of foreign PC concentrates and diagnostic reagents, significantly hinder the clinical promotion and application of these treatments. Therefore, it is imperative to urgently explore new therapeutic methods. Undoubtedly, the establishment of animal models related to the disease can greatly contribute to our understanding of its occurrence and development, as well as the exploration of novel treatment methods. In recent years, gene editing technology has made significant advancements, particularly in the CRISPR/Cas9 system. This system offers several advantages, including precise gene editing, simplicity of operation, and convenient design.

The CRISPR/Cas9 system functions by guiding the Cas9 endonuclease to cleave specific DNA double strands at the target site using the protospacer adjacent motif (PAM). Upon reinfection, the single guide RNA (sgRNA)-Cas9 complex in the CRISPR/Cas9 system binds to the target DNA sequence and cleaves the corresponding virus or plasmid (*McGinn & Marraffini, 2019*; *Jinek et al., 2012*; *Jiang & Doudna, 2017*). The cleavage-induced double-strand break damage (DSB) activates two main DNA damage repair (DDR) mechanisms: homologous recombination repair (HRR) and non-homologous end joining (NHEJ) repair. HRR allows for precise editing of target genes in the presence of homologous donor DNA templates (*Ceccaldi, Rondinelli & D'Andrea, 2016*). The CRISPR/Cas9 system has been

extensively used in various studies, including genome editing, base editing, transcriptional regulation, and epigenetic modification (*Doudna, 2020*). Moreover, it has played a crucial role in the development of mouse models for hereditary hematological disorders such as thalassemia, sickle cell disease, and hemophilia (*Huai et al., 2017*; *Li et al., 2018*; *Li et al., 2021*; *Guan et al., 2016*; *Veltrop et al., 2016*; *Tran et al., 2019*). For instance, *Huai et al. (2017)* successfully established a simulated human hemophilia B mouse model using CRISPR/Cas9 and demonstrated the efficacy of CRISPR/Cas9-mediated gene correction therapy in adult mice *in vivo* and *in vitro* cell lines. However, gene editing technology has not yet been applied to research on PCD.

In this study, we collected data on PC mutation and identified the specific mutation sites that result in the severe phenotype of hereditary PCD (*Al Harbi & El-Hattab, 2017*). Subsequently, we employed CRISPR/Cas9 gene editing technology to create heterozygous PC mutant mouse models. We then assessed the function of PC in these mouse models through genotype analysis, phenotype analysis, and PC function analysis.

## MATERIALS & METHODS

### Experimental materials
#### Animals
The experimental mice, aged 4~6 weeks, were all SPF grade C57/BL6 mice and purchased from Saiye Biotechnology Co., Ltd (Guangzhou, China). Mice were maintained and bred in specific pathogen-free conditions at the Animal Center of Central People's Hospital of Zhanjiang. Animal Ethics Committee of Central People's Hospital of Zhanjiang provided full approval for this research (approval number: ZJDY2023-54).

#### Main reagents
The Cas9 Plasmid was purchased from Vector Builder (Chicago, IL, USA); DNA Ligation Kit Ver.2.1, DH5 $\alpha$, Taq DNA polymerase and MiniBEST Universal Genomic DNA Extraction kit was acquired from Takara (Kusatsu, Japan); M2 medium, M16 medium, Poly(A) Tailing Kit, MEGA shortscript™ T7 Transcription Kit, MEGA clear kit were obtained from Thermo (Waltham, MA, USA); Phanta Master Mix was purchased from Vazyme (Jiangsu, China); Genomic DNA Kit, TIANprep Mini Plasmid Kit was acquired from Tiangen (Beijing, China), SanPrep Column DNA Gel Extraction Kit was purchased from Sango Biotec (Shanghai, China); Chloral hydrate purchased from Shanghai McLean Biochemical Technology Co., Ltd; Coagulation four test kit (PT, APTT, TT and FIB) was purchased from Rayto (Shenzhen, China); PC detection kit was purchased from Boatman Biotechnology (Shanghai, China); MOUSE PC ELISA Kit were obtained from Jingmei Bio (Hong Kong, China); Hematoxylin-eosin dye, Antigen repair solution was acquired from Wuhan Google Bio (Wuhan, China); Concentrated rabbit serumwas acquired from Haoke (Hebei, China); fibrinogen antibody was purchased from Bioss (Woburn, MA, USA); HRP conjugated Goat Anti-Rabbit IgGwere obtained from Servicebio (Wuhan, China); Dako REALTM EnVisionTM detection system were obtained from Dako (Glostrup, Denmark).

## Experimental methods
### Determining the mutation sites of PC in mouse model
Human gene mutation databases and existing case reports were used to screen for PC mutations leading to severe phenotypes, and online tools such as Polyphen-2, PROVEAN, and Mutation Taster were applied to predict the effect of selected mutation sites on PC function. Finally, the suitable mutation was determined to construct the mouse model with mutation. Clustal Omega software was performed to compare the amino acid sequence of human and mouse PC to determine the specific site of mutation on the mouse model.

### Design and synthesis of sgRNA and Donor Oligo
Two groups of sgRNAs were chosen through the online sgRNA tool CRISPR Design (https://www.synthego.com/products/bioinformatics/crispr-design-tool). One group used the classic Cas9 gene editing system while the other one applied the Cas9 D10A nickase gene editing system Design. sgRNAs with high efficiency and low off-target were screened by CCTOP (https://cctop.cos.uni-heidelberg.de/) and Cas-Offinder online tools (http://www.rgenome.net/cas-offinder/). The screened sgRNAs addition vector BpiI enzyme digestion complementary sequence was sent to GenScript Biotech Co., Ltd. (Piscataway, NJ, USA) to synthesize a pair of single stranded DNA sequences. The Donor Oligo sequence with the desired mutation was designed according to the mouse genome PC mutation site as the homologous repair template. The 140 bp homologous sequence was combined on both sides and synthesized by GenScript Biotech Co., Ltd (Piscataway, NJ, USA).

### Construction of sgRNA vector
Single-strand DNA was annealed to form double-strand DNA, which was used for subsequent connection systems to perform enzyme digestion of the carrier. A 1% agarose gel was used to detect the target band of the enzyme digestion product. After the linearized vector was recycled and purified, double-stranded DNA formed by annealing was connected to the vector. Subsequently, it was transferred into DH5$\alpha$ andcultured in LB substrate. After 14–16 h, a few monoclonal colonies were picked and then cultured for 16 h, from which we extracted plasmid. Then the recombinant plasmid was sent to Genewiz Biotech Co., Ltd. (South Plainfield, NJ, USA) for sequencing and validation.

### Transcription of sgRNA in vitro
After amplification of sgRNAs, an appropriate amount of enzyme digestion product was taken to conduct electrophoresis validation in a 1% agarose gel. The DNA Gel Extraction Kit was used to recycle the target band from the PCR product. The sequence of PCR primers is demonstrated in Table 1.

### Transcription of Cas9 in vitro
After linearization, the Cas9 plasmid was purified using gel extraction. Subsequently, the linearized Cas9 plasmid was transcribed *in vitro* to synthesize 7-methylguanine capped RNA. Finally, polyadenylate tails were added to the RNA transcribed *in vitro* by the Cas9 plasmid at the 3′ end, and the RNA was eventually purified.

**Table 1  PCR primers.**

| Primer | Primer sequence |
|---|---|
| Primer-F | GAAATTAATACGACTCACTATAGTGGAAGTAGTCAATAACCAG |
| Primer -R | AAAAAAGCACCGACTCGGTGCC |

### Microinjection and embryo transfer

Under the microscope, fertilized eggs were obtained from the uterus of donor female rats 0.5 days after mating (the donor female rats were killed by cervical dislocation), and the fertilized eggs with good shape and moderate development state were selected. Donor Oligo, sgRNA#1, and Cas9 mRNA were mixed with microinjection buffer at concentrations of 100 ng/µl, 50 ng/µl, and 100 ng/µl, respectively. The mixed RNA was extracted by microinjection and injected into the pronucleus of single-celled fertilized eggs one by one. Transfer the fertilized egg after injection to M16 medium for cultivation. During embryo transfer, the surrogate mice were anesthetized by intraperitoneal injection of 5% chloral hydrate. Cut a small incision of 1–1.5 cm in the back ovary of the surrogate mouse, and transfer the fertilized egg injected with RNA into the fallopian tube of the surrogate mother mouse. Finally, the surrogate mother mouse was placed in a clean cage and kept warm, and was put back into the cage after it woke up, waiting for the birth of the mouse.

### Genotypic identification of born mice

The toes of approximately 1-week-old mice were cut, and a mixture of 180 µl buffer GL, 20 µl Proteinase K, and 10 µl RNase A (10 mg/ml) was added and thoroughly mixed. The genomic DNA was then extracted after complete lysis of the mixture by heating in a water bath at 56 ° C. The extracted genomic DNA served as the template for PCR reaction, and the resulting product was validated through electrophoresis in a 1% agarose gel. The target product was subsequently sent to Suzhou Genewiz Biotech Co., Ltd (Suzhou, China). for sequencing. The genotypes of the mice were determined by analyzing the sequencing results. The primers used for detecting mouse mutation sites are listed in Table 2.

### Breeding for homozygous mice

The F0 generation mice that tested positive were identified through sequencing and subsequently paired with wild mice of the same age (8 weeks) for breeding purposes. Around 1 week after birth, the toes of the newborn mice were cut and sent for PCR sequencing validation. The mice that tested positive were then selected as F1 heterozygous mice. The F1 heterozygous mice were either mated with each other or with wild mice to obtain the F2 generation homozygous mice. The F2 generation homozygous mice were later mated with each other to produce more heterozygous mice.

### Detection of PC activity and antigen level

The 24 wild mice (PC (+/+)) and 20 heterozygous mice (PC (+/-)) were anesthetized by intraperitoneal injection with 5% Chloral hydrate, and blood samples were obtained from the heart. PC reagent kit (chromogenic substrate method) was used to detect, and calculate

| Table 2 | Primers designed for mutant sites detection in mice. |
|---|---|
| **Primer name** | **Primer sequence** |
| Mouse Proc (G399S)-F | GGAGATCCTCGTCCACCCTAACTACAC |
| Mouse Proc (G399S)-R | TGTCCTCTGTCTTGGTACTCACTAGCCC |
| Sequencing primer | ACATTGCTCTGCTCCGCCTAG |

the PC's activity of the blood based to the standard curve. The antigen level of PC was detected by ELISA kit of mouse PC and calculated according to the standard curve.

### Blood routine and four coagulation tests

After completing the detection of PC activity and antigen level, the remaining blood samples were subjected to a blood routine and four coagulation tests. Blood routine testing with an automatic blood analyzer. The prothrombin time (PT), activated partial thromboplastin time (APTT) and thrombin time (TT) were detected with the automatic coagulation analyzer, and the fibrinogen content (FIB) was calculated.

### Histopathological examination

Wild mice and heterozygous mice from the same litter were euthanized by cervical dislocation, whose liver, spleen, and kidneys were taken for H&E staining and immunohistochemistry.

### Statistical methods

GraphPad Prism 9 software was used for graphical and statistical analyses. The blood routine and coagulation results were represented by ($\bar{x}\pm$s), and the activity and antigen level of PC were tested using an independent sample unpaired $T$-test method. The difference was statistically significant when $P < 0.05$ was used.

## RESULTS

### Clinically novel mutations in PC were discovered

A recent case report highlighted a novel mutation in the PROC gene (*Al Harbi & El-Hattab, 2017*). The report described an infant who developed multiple lumps in the lower limbs two months after birth. The condition worsened after undergoing a muscle biopsy of the right thigh at five months, leading to fulminant purpura. The infant also experienced anemia, thrombocytopenia, consumptive coagulopathy (prolonged PT, APTT, and low fibrinogen), and fibrinolysis (increased D-dimer), which were consistent with disseminated intravascular coagulation (DIC). Further analysis revealed that the PC level was significantly low (less than 10%, normal range: 68–143%), indicating autosomal recessive PCD. The patients were confirmed to have a new homozygous missense mutation PROC c.1198G>A (p.Gly400Ser) through PROC gene sequencing. However, the mutation site was not extensively studied in this case report, and no related reports have been published.

### Functional prediction of PROC mutation

Next, to understand the function of novel mutation, we used software along with clinical data to predict. The results of PROVEAN, PolyPhen-2 and Mutation Taster showed that

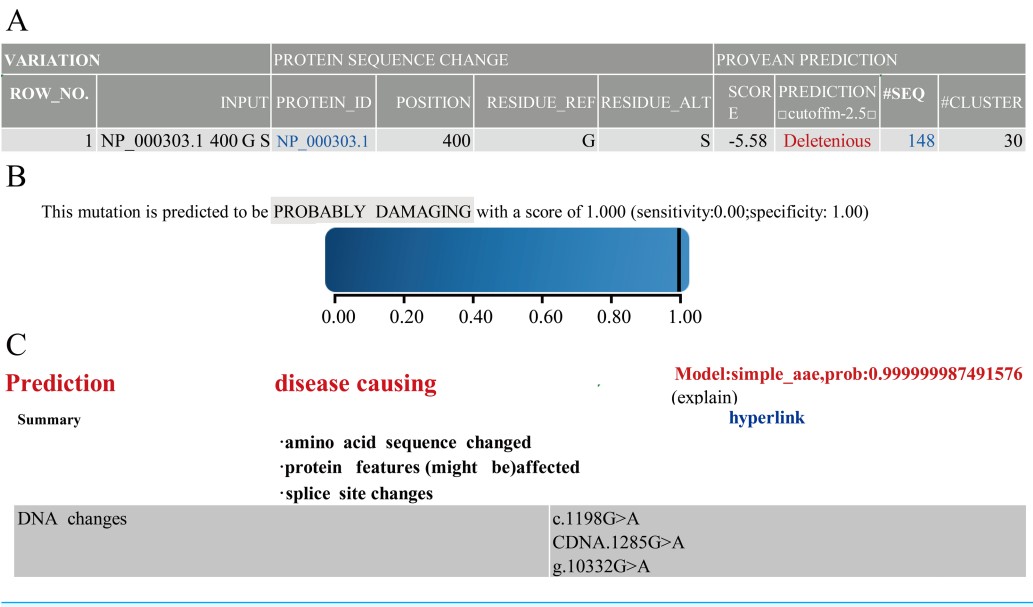

A

| VARIATION | | PROTEIN SEQUENCE CHANGE | | | | PROVEAN PREDICTION | | | |
| --- | --- | --- | --- | --- | --- | --- | --- | --- | --- |
| ROW_NO. | INPUT | PROTEIN_ID | POSITION | RESIDUE_REF | RESIDUE_ALT | SCORE | PREDICTION □cutoffm-2.5□ | #SEQ | #CLUSTER |
| 1 NP_000303.1 400 G S | | NP_000303.1 | 400 | G | S | -5.58 | Deleterious | 148 | 30 |

B

This mutation is predicted to be PROBABLY DAMAGING with a score of 1.000 (sensitivity:0.00;specificity: 1.00)

0.00   0.20   0.40   0.60   0.80   1.00

C

**Prediction**          **disease causing**                  .              **Model:simple_aae,prob:0.999999987491576**
                                                                             (explain)
   Summary                                                                   **hyperlink**

                                ·amino acid sequence changed
                                ·protein features (might be)affected
                                ·splice site changes

| DNA changes | c.1198G>A CDNA.1285G>A g.10332G>A |
| --- | --- |

**Figure 1   Prediction results of PROVEAN.** Polyphen-2 (A) Mutation Taster (B) and Prediction results (C) of software.

PROC c.1198G>A(p.Gly400Ser) caused PC to be seriously dysfunctional. Based on clinical data and results of prediction software (Fig. 1), PROC c.1198G>A (p.Gly400Ser) was selected as the target for the mutant mouse model.

## Comparison of PC amino acid sequence between human and mouse
After predicting the function and determining the target mutation, we compared the difference in PC amino acid sequence between humans and mice. The results showed that glycine at human site 400 is homologous to serine at mouse site 399 (Fig. 2). Then, we investigated the mouse Proc gene (NM_001042767.3) in NCBI and learned that the serine at site 399 is located in exon 9. Thereby we chose the target site to design sgRNAs and Donor Oligo and introduced the mutation site through HRR.

## Screening of sgRNA
Having a knowledge of the relative information of the novel mutation, we designed the candidate sgRNAs for follow-up experiments. The CRISPRater score predicted by CCTop software for candidate sgRNAs was 0.64 MEDIUM (sgRNA#1), 0.67 MEDIUM (sgRNA# 2), and 0.86 HIGH (sgRNA#3), respectively, While the Cas Offerer fast universal algorithm showed that sgRNA#1 might be the most suitable one with the least potential off-target sequences (Fig. 3). (sgRNA1: GACACGAGAGAGACCTGCTGTGATGG. TGG is the PAM sequence).

## Construction and validation of sgRNA vector
After the sgRNA was selected, we designed and synthesized sgRNA#1 with the target sequence and connected it with the plasmid vector pRP[CRISPR]-hCas9, named pRP[CRISPR] -HCas9-U6 >{GACACGAGAGACGCCTGTGA} (Fig. 4A). The

```
. Human   LNFIKIPVVPHNECSEVMSNMVSENMLCAGILGDRQDACE[G]DSGGPMVASFHGTWFLVGL   419
. Mouse   LTFIRIPLVARNECVEVMKNVVSENMLCAGIIGDTRDACD[G]DSGGPMVVFFRGTWFLVGL   418
          *.**:**:* :*** ***.*:**********:** :***:********. *:********
```

**Figure 2  Comparison of PC amino acid sequences between humans and mice.** "*" indicates that the amino acid sequence is completely consistent; ":" indicate that the amino acid sequence is highly conserved; "." indicates that the amino acid sequence is conserved; no marker indicates that the amino acid sequence is not conserved and the red box indicates the target site.

**A**

Summary

| Target Sequence | Bulge Type | Bulge Size | Mismatch | Number of Found Targets |
|---|---|---|---|---|
| GACACGAGAGACGCCTGTGANGG | × | 0 | 0 | 1 |
| GACACGAGAGACGCCTGTGANGG | × | 0 | 3 | 2 |
| GACACGAGAGACGCCTGTGANGG | × | 0 | 4 | 63 |

**B**

Summary

| Target Sequence | Bulge Type | Bulge Size | Mismatch | Number of Found Targets |
|---|---|---|---|---|
| CCCCCCACTGTCACCATCACNGG | × | 0 | 0 | 1 |
| CCCCCCACTGTCACCATCACNGG | × | 0 | 2 | 3 |
| CCCCCCACTGTCACCATCACNGG | × | 0 | 3 | 43 |
| CCCCCCACTGTCACCATCACNGG | × | 0 | 4 | 183 |

**C**

Summary

| Target Sequence | Bulge Type | Bulge Size | Mismatch | Number of Found Targets |
|---|---|---|---|---|
| TGACAGTGGGGGGCCCATGGNGG | × | 0 | 0 | 1 |
| TGACAGTGGGGGGCCCATGGNGG | × | 0 | 2 | 1 |
| TGACAGTGGGGGGCCCATGGNGG | × | 0 | 3 | 11 |
| TGACAGTGGGGGGCCCATGGNGG | × | 0 | 4 | 139 |

**Figure 3  Cas-Offinder analysis of sgRNA#1 (A), sgRNA#2 (B) and sgRNA#3 (C).**

recombinant plasmid was sequenced for validation, and the sequence was shown in gray (Fig. 4B), indicating the successful construction of the sgRNA vector, which was *in vitro* transcribed for subsequent embryonic injection.

## Synthesis of Donor oligo

We next designed and synthesized the donor oligo with the target site: G399s (GGT to AGC) and sequence:(AGC is introduced for the purpose of mutation): GTGGTCTCGGAGAACATGCTGTGTGCAGGCATCATTGGGGACACGAGAGACG CCTGTGATAGCGACAGTGGGGGGCCCATGGTGGTCTTCTTTCGGGGTACCTG GTTCCTGGTGGGCCTGGTGAGCTGGGGTGAGGGCTGTGG.

## Genotype identification of born mice

After transferring the embryo, we identified two F0 generation positive mice (8 and 47) and all F0 generation mice were bred with wild-type mice to obtain F1 generation mice. Then, we obtained and identified six F1 generation positive mice (4, 10, 13, 15, 16 and 17). Subsequently, F1 generation mice were bred with each other to obtain F2 generation mice,

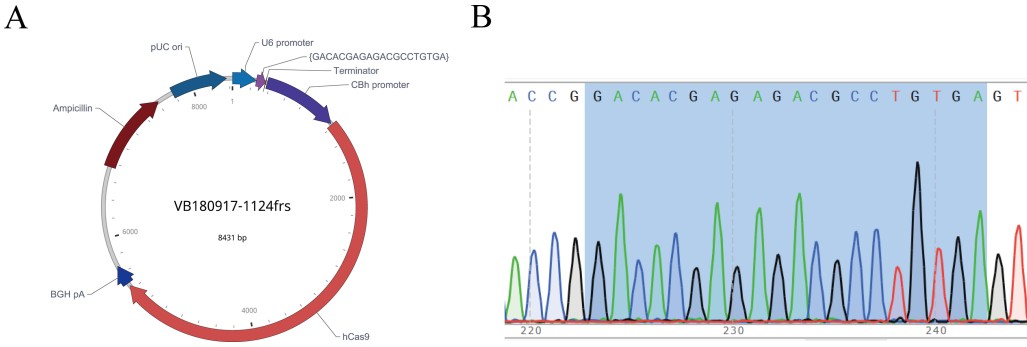

**Figure 4** **Map of recombinant plasmid (A) and sequencing results (B).** In the sequencing result graph, different colored sequencing peaks represent different base sequences, specifically red representing T, black representing G, green representing A and blue representing C. The base sequences corresponding to the sequencing peak are listed above the peak.

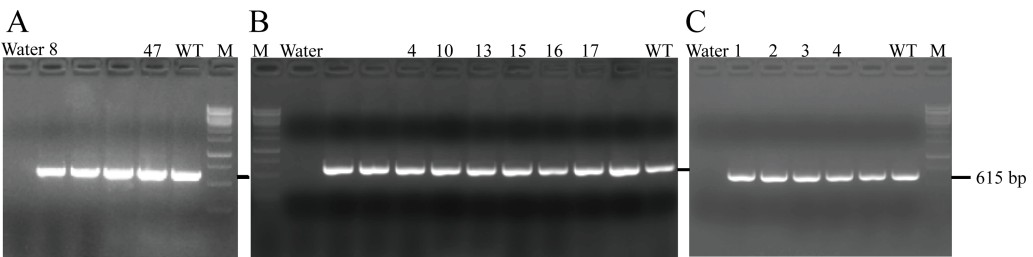

**Figure 5** **PCR results of F0 (A), F1 (B) and F2 (C) generation mice.**

and eleven positive mice (1, 2, 3, 4, 6, 8, 9, 10, 11, 12, 13) were obtained after identification. PCR results of F0, F1, and F2 generation mice were exhibited in Figs. 5A–5C, respectively, with a target band of 615 bp. The sequencing results of mice numbered 8, 4, and 1 for F0, F1, and F2 were shown in Figs. 6A–6C, respectively. The sequence marked in gray represented the target site G399S (GGT to AGC). Furthermore, the sequencing results showed that the two F0 generation positive mice were chimeras, while the F1 and F2 generation positive mice were heterozygous. All in all, there were 20 mice in total, while no homozygous was obtained. We further propagated mice and conducted genotype identification on a total of 153 mice, but no homozygotes were finally obtained.

## Phenotype and anatomical results of heterozygous mice

There were 44 heterozygous and wild mice in F1 and F2 in total, and up to now no homozygous mice have been obtained. Among the heterozygous mice, no manifestations of spontaneous bleeding or thrombosis were observed on the body surface, and some mice experienced blindness (Fig. 7A). In addition, we carried out an autopsy and the results showed that one end or the whole length of the spleen of some heterozygous mice was congested with dark purple (Fig. 7B). In the process of mouse breeding, an F1-30 pregnant female mouse was found to have miscarried and died (Fig. 7C). Observation of the body

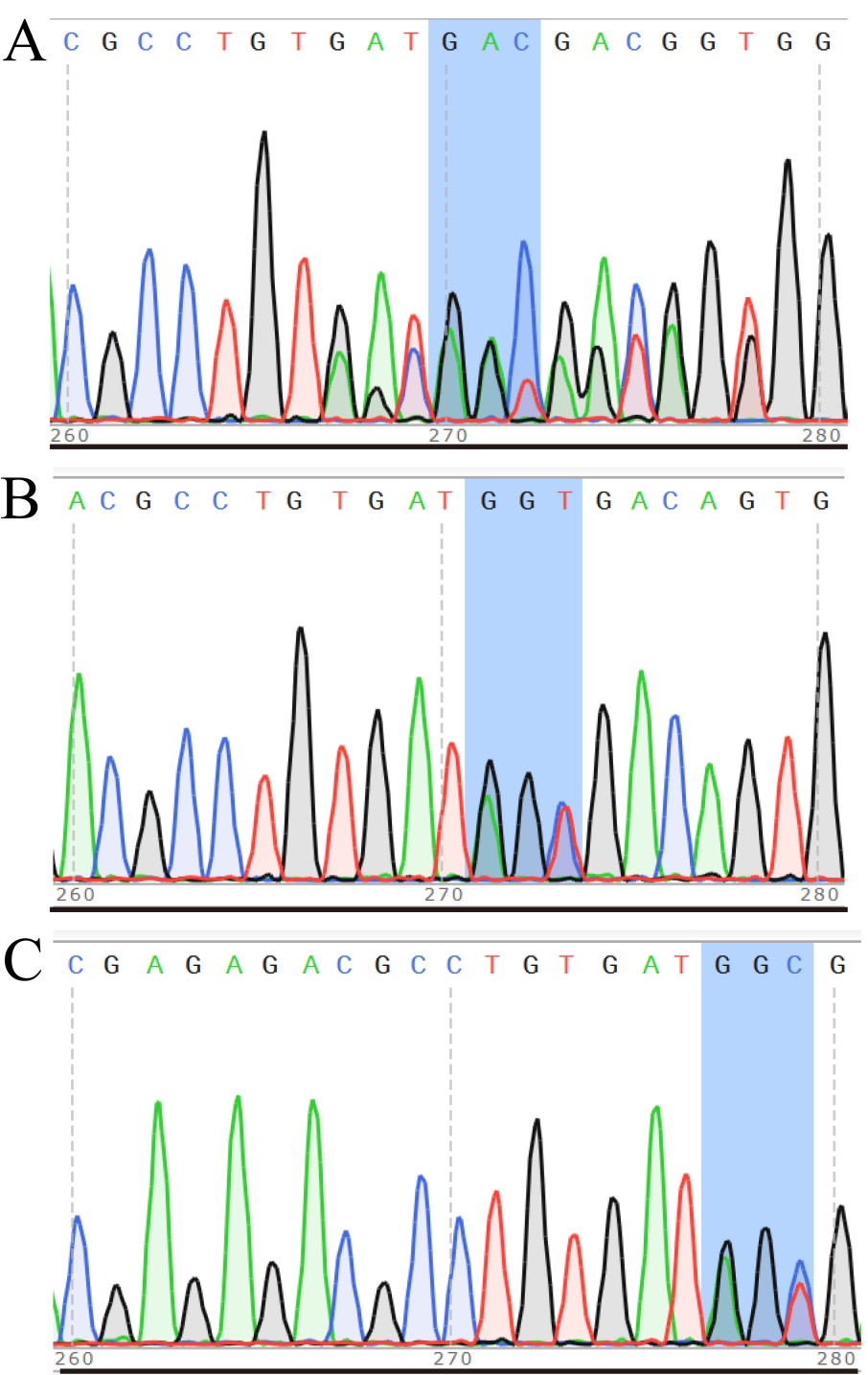

**Figure 6  Sequencing results of some mice in F0 (A), F1 (B) and F2 (C) generations.** In the sequencing result graph, different colored sequencing peaks represent different base sequences, specifically red representing T, black representing G, green representing A and blue representing C. The base sequences corresponding to the sequencing peak are listed above the peak.

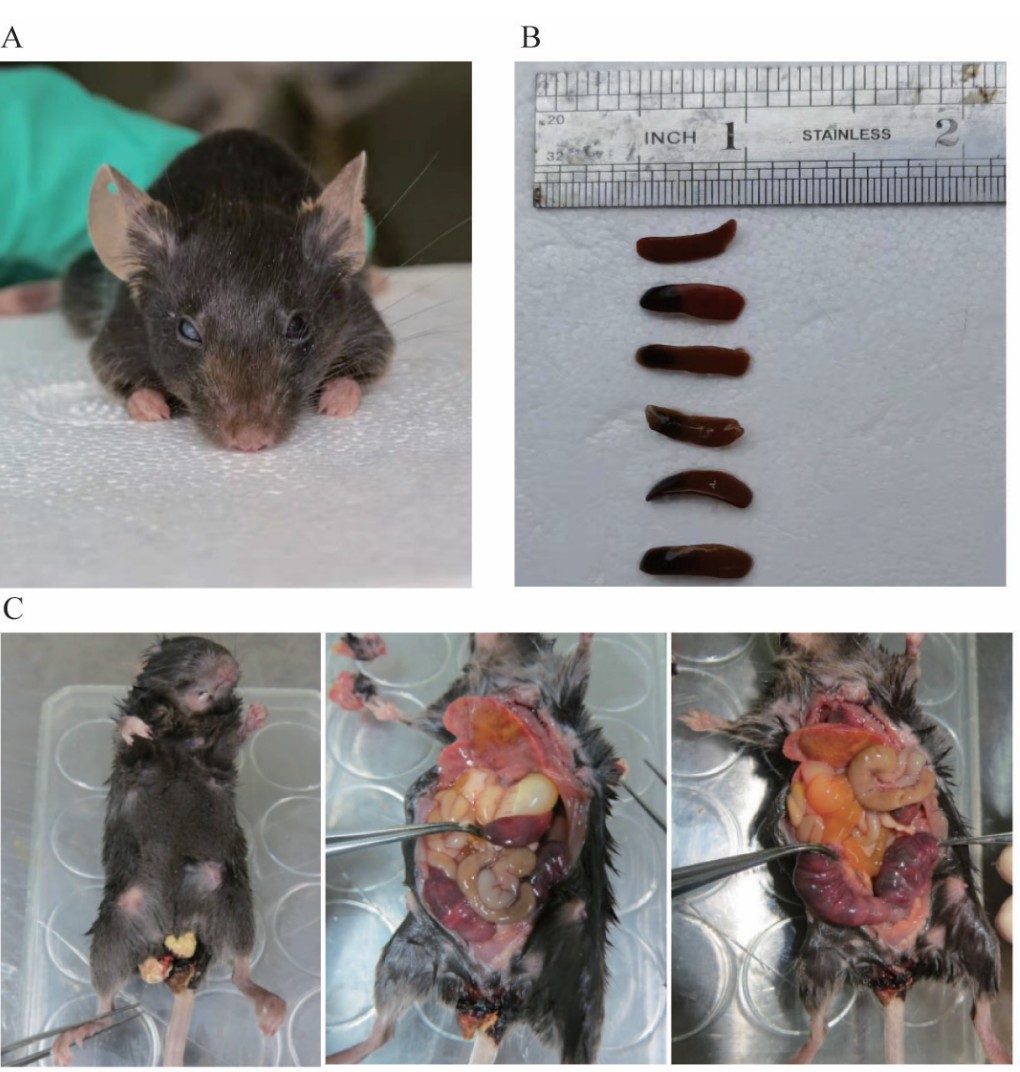

**Figure 7** (A) shows blindness in the right eye of mice. (B) shows the spleen of different type of mouse. The first is the spleen of a wild mouse, and the last five are the spleens of heterozygous mice with congestion at one end. (C) shows a dead pregnant female rat.

surface revealed that the female mouse had a larger abdomen, protruding and red nipples, and there were signs of bleeding near the vagina without trauma. After the autopsy, we observed that the female mouse's heart and lungs were congested, the liver was mottled, the stomach was filled with water, the front end of the spleen was congested, the intestines and kidneys were relatively normal, and both uterine horns were enlarged, containing multiple embryos that had not reached the age of production. We then attempted to separate embryos but failed. Since the mice died within 12 h, it is speculated that the cause of death was miscarriage.

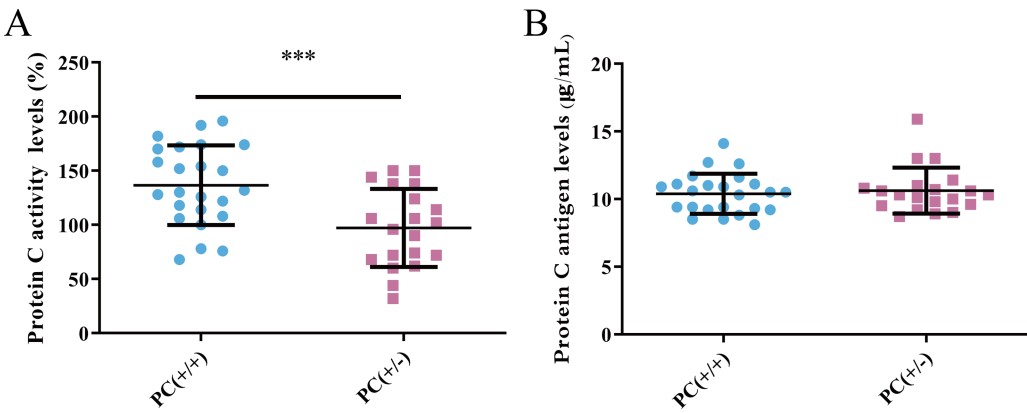

**Figure 8** Activity (A) and antigen levels (B) of PC.

## Detection of PC activity and antigen level

PC activity and antigen levels were examined in the study. The average activity of PC in heterozygous mice was found to be 97.1% according to the standard curve, whereas in wild mice, it was 136.7%. This difference was statistically significant ($P < 0.001$) (Fig. 8A). The average level of PC antigen in heterozygous mice was found to be 10.6 µg/ml, and in wild mice, it was 10.4 µg/ml. This difference was not statistically significant ($P > 0.05$) (Fig. 8B). These findings suggest the presence of PCD type II.

## Results of blood routine and four coagulation tests

To determine whether the mutation exerts effects on the activities of blood, we collected the mice's blood and found there was no significant difference in blood routine between heterozygous mice and wild mice ($P > 0.05$) (Table 3). However, the results of the four coagulation tests (Table 4) showed that the plasma of the hybrid mice had no coagulation reaction, the PT, APTT and TT prolonged, while the fibrinogen content reduced, suggesting that the secondary consumptive coagulopathy was consistent with the clinical results.

## H&E staining and immunohistochemistry

Through H&E staining assay (Fig. 9), we found that the livers of the heterozygous mice in the PC (+/-) group showed steatosis and hydropic degeneration, and there were a large of lymphocytes and plasma cell infiltration near the hepatic sinuses. Necrotizing and shedding epithelial cells and cell fragments were observed in the lumen of the renal tubules, forming cellular or granular tubular type. Additionally, the spleen showed deposition of hemosiderin, and the PC (+/-) group demonstrated a diffuse distribution of brown-yellow positive expression (Fig. 10).

## DISCUSSION

CRISPR/Cas9, a simple and efficient gene editing method, has been widely utilized for constructing animal disease models. This technique has provided valuable empirical guidance for creating PC mutant mouse models (*Huai et al., 2017*; *Li et al., 2018*; *Li et al.,*

**Table 3  Results of routine blood tests.**

| Project | PC (+/+) group | PC (+/-) group |
|---|---|---|
| Prythrocyte ($10^{12}$/L) | $7.42 \pm 0.35$ | $7.22 \pm 0.38$ |
| Hemoglobin concentration(g/L) | $117.67 \pm 8.96$ | $119.21 \pm 8.52$ |
| Peukocyte ($10^9$/L) | $1.42 \pm 0.58$ | $1.54 \pm 0.95$ |
| Platelet ($10^9$/L) | $473.20 \pm 35.22$ | $513.79 \pm 132.18$ |
| Lymphocyte percentage(%) | $56.32 \pm 9.86$ | $51.83 \pm 13.28$ |
| Monocyte percentage(%) | $17.12 \pm 2.31$ | $15.68 \pm 5.42$ |
| Eosinophil percentage(%) | $0.70 \pm 0.54$ | $0.69 \pm 0.84$ |
| Basophil percentage(%) | $0.72 \pm 0.33$ | $0.63 \pm 0.48$ |
| Neutrophil percentage(%) | $25.14 \pm 9.95$ | $31.17 \pm 12.83$ |

**Table 4  Results of the four coagulation items.**

| Project | PC(+/+) group | PC(+/-) group |
|---|---|---|
| PT(sec) | $10.5 \pm 4.3$ | – |
| APTT(sec) | $27.8 \pm 7.6$ | – |
| TT(sec) | $30.3 \pm 4.5$ | – |
| FIB(g/L) | $1.2 \pm 0.2$ | – |

**Notes.**
–, indicates that there is no solidification reaction, and the value cannot be obtained.

*2021*; *Guan et al., 2016*; *Veltrop et al., 2016*; *Tran et al., 2019*). In this study, we introduced target mutations using the homologous end junction repair mode of the CRISPR/Cas9 system, resulting in the construction of a heterozygous mouse model with PC mutation. Our findings contribute to a better understanding of the specific mechanisms underlying this new mutation in PC, which in turn facilitates comprehension of disease occurrence and development.

The clinical manifestations of hereditary PCD exhibit a wide range of symptoms, and the relationship between genotype and phenotype remains unclear (*Lu et al., 2019a*). While most patients with homozygous PCD typically develop purpura fulminans shortly after birth, the reported case presented with delayed purpura fulminans (*Al Harbi & El-Hattab, 2017*). Since there are no homozygous mice available, it is currently impossible to determine if they would also exhibit similar phenotypes. Additionally, the genotypes of all the mice born so far do not follow the expected Mendelian distribution, and the potential impact of the mutated gene on mouse development and survival cannot be ruled out. Our next step is to increase the number of mouse breeding attempts in order to obtain homozygous mice. Subsequently, we will combine the blood routine, coagulation results, PC content and enzyme activity, H&E staining and immunohistochemistry data of normal mice, heterozygous mice and homozygous mice to analyze the relationship between genotype, phenotype and PC function.

In this study, heterozygous mice did not exhibit any spontaneous bleeding phenotype. In clinical practice, most heterozygous carriers are asymptomatic, although the risk of thrombosis is significantly increased. The appearance of the phenotype is also influenced

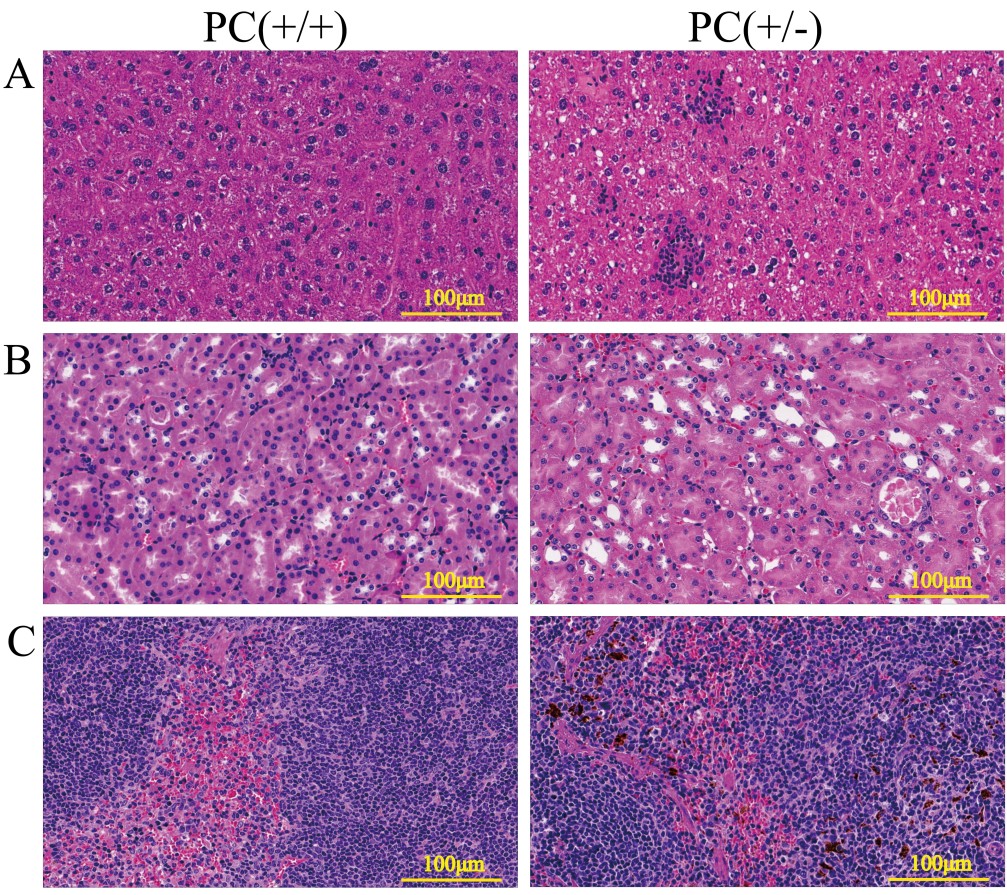

PC(+/+)      PC(+/-)

**Figure 9** HE staining of liver (A), kidney (B) and spleen (C), ×200, scale 100μm.

by other genetic and acquired factors (*Soria et al., 1995*; *Ding et al., 2015*). Some mice died during reproduction due to dysgenesis, resulting in excessive thinness, and two mice experienced blindness (Fig. 7A). Eye lesions have also been observed in children with severe congenital PCD (*Tang et al., 2022*). While interbreeding F1 hybrid mice to obtain homozygous mice, one pregnant mother mouse died, and it was not possible to determine the genotype of the embryo due to the inability to separate it (Fig. 7C). The mortality of the mother mouse may be attributed to miscarriage caused by intrauterine mortality of homozygous mice or other factors, such as thrombosis resulting from pregnancy risk factors in heterozygous mother mice. Considering the risk of miscarriage and stillbirth in females with PCD, as well as the fact that the mother mouse died within 12 h, it is speculated that the cause of death may be miscarriage (*Sanson et al., 1996*).

Hereditary PCD is a disease caused by mutations in the PROC gene. Most mutations result in type I defects, characterized by a decrease in both PC activity and antigen levels. Type II defects, which occur in 10 to 15% of cases, are characterized by normal antigen levels but reduced PC activity (*Mackie et al., 2013*). The detection of PC activity and antigen levels revealed that heterozygous mice had significantly lower PC activity compared to wild

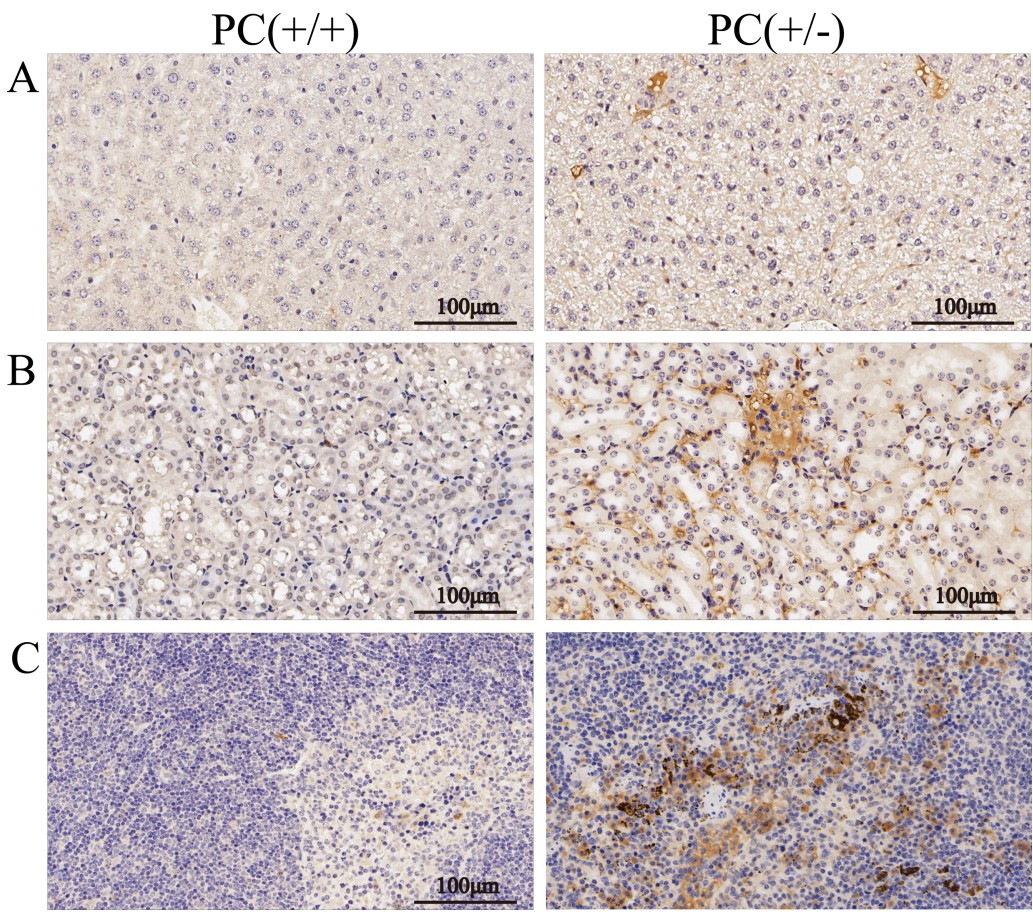

**Figure 10   Immunohistochemistry of liver (A), kidney (B) and spleen (C), ×200, scale 100 μm.**

mice ($P < 0.001$), but there was no significant difference in antigen levels ($P > 0.05$) (Fig. 8), indicating a type II PCD, consistent with the clinical data of the pediatric patient (*Al Harbi & El-Hattab, 2017*). There were no statistically significant differences in blood routine results between heterozygous mice and wild mice ($P > 0.05$) (Table 3). The results of the four coagulation tests showed that the plasma of hybrid mice did not exhibit coagulation reactions within 120 s, leading to the instrument's inability to detect values. However, the PT, APTT, and TT were prolonged, and the fibrinogen content was reduced (Table 4), suggesting secondary consumption coagulopathy. Similarly, clinical practice has shown that patients carrying homozygous mutations PROC c.1198G>A (p.Gly400Ser) experience coagulation disease and fibrinolysis (increased D-dimer), which manifests as disseminated intravascular coagulation (*Al Harbi & El-Hattab, 2017*). The liver is the primary organ responsible for producing human PC, and it is also one of the organs most affected by PCD. H&E staining results revealed that the livers of heterozygous mice in the PC (+/-) group exhibited steatosis and hydropic degeneration. Additionally, there was a significant infiltration of lymphocytes and plasma cells near the hepatic sinuses (Fig. 9A). Renal tubules in these mice showed necrotizing and shedding epithelial cells and

cell fragments, forming a cellular or granular tubular type (Fig. 9B). This could potentially be associated with impaired cellular protective function of PC (*Lu et al., 2019b*). Some studies have also reported that patients with PCD suffer from immunoglobulin A nephropathy (*Ikehata et al., 2022*). Furthermore, there have been cases of PCD where patients exhibited bronchi and alveoli rich in macrophages containing hemosiderin, leading to a diagnosis of pulmonary hemosiderosis (*Ajmi et al., 2023*). Similarly, we observed deposition of hemosiderin in the spleen of mice, indicating splenic hemorrhage. Dissection of the spleen revealed congestion at one end or along its entire length, with a dark purple colour (Fig. 9C). Immunohistochemistry of the PC (+/-) group demonstrated a diffuse distribution of brown-yellow positive expression, indicating significant fibrin deposition in the liver, spleen, and kidneys of the heterozygous mice (Fig. 10). Human APC has been shown to increase fibrinolysis in a dose-dependent manner (*Bertina et al., 1988*). Therefore, the decrease in PC activity observed in the PC (+/-) group may contribute to the appearance of fibrinogen deposition in multiple organs of the mice. Additionally, it has been suggested that blood clots may resist fibrinolysis (*Foley, Ferris & Brummel-Ziedins, 2012*).

## CONCLUSIONS

We successfully introduced the target mutation of PC mutation into the mouse model using the homologous end junction repair mode of the CRISPR/Cas9 system. As a result, we obtained heterozygotes of the mouse model of PC mutation. While there were no visible phenotypes in the heterozygous mouse model, except for blindness and abortion, we observed significant differences in coagulation parameters, PC activity levels, H&E staining and immunohistochemistry results of the liver, spleen, and kidney compared to healthy mice. In conclusion, we partially validated the function of PC in the mouse model through genotype, phenotype, and PC function analysis. Our next step is to increase the number of mice bred to obtain homozygous mice, combine heterozygous data, and analyze their genotype, phenotype, and PC function.

### Funding
This work was supported by the National Natural Science Foundation of China (No. 81971886, No. 82170053 and No. 82370099), the Zhujiang Talent Program (No. 2019QN01Y279), the Zhanjiang Science and Technology Project (No. 2021A05137, No. 2022A01107 and No. 2022A01075) and the Discipline Construction Fund of Zhanjiang Central Hospital, Guangdong Medical University (No. 2022A03). The funders had no role in study design, data collection and analysis, decision to publish, or preparation of the manuscript.

### Grant Disclosures
The following grant information was disclosed by the authors:
National Natural Science Foundation of China: 81971886, 82170053, 82370099.

Zhujiang Talent Program: 2019QN01Y279.
Zhanjiang Science and Technology: 2021A05137, 2022A01107, 2022A01075.
Discipline Construction Fund of Zhanjiang Central Hospital, Guangdong Medical University: 2022A03.

## Competing Interests

The authors declare there are no competing interests.

## Author Contributions

- Ya Liu performed the experiments, authored or reviewed drafts of the article, and approved the final draft.
- Maoping Cai analyzed the data, authored or reviewed drafts of the article, and approved the final draft.
- Yan Chen performed the experiments, prepared figures and/or tables, and approved the final draft.
- Guocai Wu analyzed the data, prepared figures and/or tables, and approved the final draft.
- Songyu Li performed the experiments, authored or reviewed drafts of the article, and approved the final draft.
- Zhanghui Chen conceived and designed the experiments, authored or reviewed drafts of the article, and approved the final draft.

## Animal Ethics

The following information was supplied relating to ethical approvals (i.e., approving body and any reference numbers):

Animal Ethics Committee of Central People's Hospital of Zhanjiang provided full approval for this research (approval number: ZJDY2023-54)

## Data Availability

The raw data is available in the Supplementary File.

## Supplemental Information

Supplemental information for this article can be found online at http://dx.doi.org/10.7717/peerj.17261#supplemental-information.

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
