# Peer review of "Validation for the function of protein C in mouse models"

_PeerJ, doi:10.7717/peerj.17261_

## Round 0.1 · original submission · Major Revisions

Dear Authors,

As per the comments received from our expert reviewers, the manuscript has some points to be addressed. Please make corrections and resubmit ASAP.
All the best

**Language Note:** PeerJ staff have identified that the English language needs to be improved. When you prepare your next revision, please either (i) have a colleague who is proficient in English and familiar with the subject matter review your manuscript, or (ii) contact a professional editing service to review your manuscript. PeerJ can provide language editing services - you can contact us at copyediting@peerj.com for pricing (be sure to provide your manuscript number and title). – PeerJ Staff

Reviewer 1 ·

Basic reporting

The primary weakness of the reporting here is prematurely overstating the conclusions regarding successfully establishing a physiologically accurate mouse model of hereditary PC deficiency. The current data provided does not substantiate these conclusory claims. This is because authors have thus far only obtained heterozygous mice and have not proven a successful generation of homozygotes exhibiting a severe deficiency phenotype.
Additionally, the authors have failed to provide critical details on the specific human PC variant identified, including any clinical or functional evidence directly linking this mutation to a severity consistent with the mouse phenotype predictions. Asserting this variant causes loss of PC activity based purely on computational algorithms is insufficient evidence.

Experimental design

The lack of homozygous mice means the severe PC deficiency phenotype predicted to be caused by this mutation has not been evaluated or confirmed. Failing to achieve the genotype necessary to test key assumptions and predictions of the model represents a major design flaw.
Suggested improvement: The authors need to continue efforts to generate viable homozygous mice to complete intended genotype-phenotype assessments. Troubleshooting assistance may be required if technical barriers are faced.

No data demonstrates the introduced variant impairs PC activity itself. The assumption it causes dysfunction is based solely on computational algorithms. Quantitatively measuring and comparing PC levels between wildtype, heterozygous, and eventual homozygous mice is essential to confirming the pathogenic effects of this mutation.
Suggested improvement: Incorporate assays quantifying PC activity reduction resulting specifically from the introduced variant. Compare function between mice of all three genotypes.

The human mutation was selected to model severe, early-onset PC deficiency. However clinical details verifying this specific variant causes severe patient thrombosis is lacking. The rationale for choosing and predictions made about this variant are thus questionable in the absence of confirmatory patient data.
Suggested improvement: Provide sources describing clinical manifestations linked to this variant, including evidence it matches the phenotype intended to be modeled in mice. Functional evidence from patients is ideal.

Validity of the findings

While the present data represents initial steps toward the intended disease model, the lack of critical genotype and functional validation experiments means the concrete validity of the model is still unknown. Yet, the conclusions state such a validated model has already been successfully created. This inappropriate extrapolation from the data provided undermines the findings' overall validity.

Suggested improvements would be:

Avoid definitively concluding a robust, validated model exists until data from homozygotes is available.
Incorporate quantitative assays in mice of all three potential genotypes to directly determine the introduced mutation's effects on PC activity.
Repeatedly acknowledged data is thus far only from heterozygotes, and precisely what validation is still needed in homozygotes. A transparent discussion of limitations is warranted.

Additional comments

Abstract:
Overall good summary of study aims, methods, key results and conclusions.

Lines 20-21: The specific mutation site could be stated clearly here for clarity.

Lines 39-42: Conclusions may be too strong based on data so far, as only heterozygotes were obtained. Suggest toning down the claim of the robust model establishment until homozygotes are evaluated.

Introduction:

Lines 44-51: Provide more background on genetics and pathogenesis of PC deficiency here to set context, especially details on autosomal recessive severe cases.

Lines 52-64: Expand on the importance of animal models for understanding disease and testing new treatments.

Results:

Lines 203-209: Provide key details of clinical severity seen with this variant to justify choice. Were functional studies done?

Lines 270-277: Improve paragraph clarity on PC activity differences between genotypes. Statistics can be presented more clearly.

Lines 284-291: Reduce redundant statements. Combine paragraphs for better flow.

Discussion:

Lines 294-307: Expand the limitations section significantly. Discuss inability to obtain homozygotes yet and what is still needed to validate the model based on predictions fully.

Lines 308-316: Paragraph on CRISPR has good background but is tangential. Consider shortening or moving to introduction.

Lines 357-360: Conclusions and future direction should reflect limitations of heterozygote-only data so far. Model not rigorously confirmed.

Reviewer 2 ·

Basic reporting

The main flaw in this reporting is that it overstates the results of the establishment of a physiologically realistic mouse model of hereditary PC deficiency, which was accomplished too soon. These conclusory claims are not supported by the available data currently. This is since researchers have only produced heterozygous mice thus far and have not demonstrated the ability to produce homozygotes that successfully display a severe deficiency phenotype.
Furthermore, the authors have not included any clinical or functional evidence that directly links the identified human PC variant to a severity that is consistent with the mouse phenotype predictions. There is not enough data to conclude that this variant reduces PC activity based only on computational algorithms.

Experimental design

The severe PC deficiency phenotype that is thought to be brought on by this mutation has not been examined or verified because there are no homozygous mice. A significant flaw in the design is the inability to obtain the genotype required to test important hypotheses and predictions of the model.
Recommendation for improvement: In order to finish the planned genotype-phenotype analyses, the authors must keep working to produce viable homozygous mice. Should technical obstacles arise, troubleshooting support might be needed.
There is no evidence that the introduced variant hinders PC activity in and of itself. The only basis for the assumption that it causes dysfunction is computational algorithms. Confirming the pathogenic effects of this mutation requires quantitatively measuring and comparing PC levels between wildtype, heterozygous, and eventually homozygous mice.
Assays measuring the decrease in PC activity that is specifically caused by the introduced variant should be included. This is a suggested improvement. Compare the three genotypes of mice's functions.
To simulate a severe, early-onset PC deficiency, the human mutation was chosen. However, there are insufficient clinical data to confirm that this particular variation results in severe patient thrombosis. In the absence of corroborating patient data, the reasoning behind the selection and forecasts regarding this variant are therefore dubious.
Improvement recommendation: Provide references to publications that detail the clinical symptoms associated with this variant, along with proof that it corresponds to the mouse phenotype that is meant to be modelled. Patient-provided functional evidence is ideal.

Validity of the findings

Although the current data indicates a first step towards the intended disease model, the model's actual validity is still unknown due to the absence of crucial genotype and functional validation experiments. However, the findings indicate that a successfully validated model has already been developed. This unwarranted extrapolation from the given data compromises the general validity of the findings.

Proposed enhancements include:
Do not declare with certainty that a solid, validated model exists until homozygotes' data is in.
Include quantitative assays in mice with each of the three possible genotypes to ascertain the impact of the introduced mutation on PC activity directly.
Consistently acknowledged data only comes from heterozygotes thus far, and it is unclear exactly what validation is still required in homozygotes. It is necessary to have an open dialogue about the limitations.

Additional comments

• Because only heterozygotes were found, conclusions drawn from the data thus far may be overly strong. Reduce the asserted strength of the robust model establishment until homozygotes have been assessed.
• In introduction: To provide context, expand on the genetics and pathophysiology of PC deficiency in this section. Pay particular attention to the information on autosomal recessive severe cases. Describe the significance of using animal models to study disease and test novel treatments.
• In results: Make the paragraph about the variations in PC activity amongst genotypes clearer. It is possible to present statistics more clearly.
• In discussion: Extend the section on limitations substantially. Talk about the fact that homozygotes have not yet been obtained and what is still required to fully validate the model based on predictions.
• In conclusion: Conclusions and future planning should take into account the limitations of the available heterozygote-only data. Model not thoroughly verified.

---

## Round 0.2 · accepted · Accept

Dear Authors,

With pleasure, I inform that the manuscript is accepted for publication. This is an editorial acceptance and still need few more tasks to be completed before publication. Therefore, I request you to be available for few days to avoid any further delays.

All the best for your future submissions.

Reviewer 1 ·

Basic reporting

The authors have satisfactorily addressed most of the major concerns raised in the previous round of reviews, but still few text corrections remain to be done before publication.
1. Explicitly state in the abstract and introduction that homozygous mice could not be obtained and analyzed in the current study.
2. Explain more specifically in the discussion how the coagulation, histological, and immunohistochemical data from heterozygotes does suggest an altered phenotype, albeit milder than expected for homozygotes. Further interpret the implications of this data for the utility and limitations of heterozygotes.
3. Touch upon in the conclusion whether breeding challenges may indicate some degree of homozygous embryonic lethality and how this would impact the mouse model overall. Mention any plans to investigate this possibility.

Experimental design

Despite these remaining points, the authors have been responsive to the critiques, and the revised manuscript is significantly improved.

Validity of the findings

Satisfied with corrections done

Additional comments

While not yet definitive, this work represents a solid step toward an important goal and sets the stage for continued progress in this area.

Reviewer 2 ·

Basic reporting

it is clear enough to understand.

Experimental design

Changes which were recommended has been incorporated well.

Validity of the findings

Authors have put the efforts to incorporate the changes suggested in earlier review.